# OpenReview forum: "Spectral-Topological Phase Transitions in Bayesian Neural Networks: Persistent Homology Meets Uncertainty Quantification"
_mathai.club/MathAI/2026/Conference — Submitted to 2026_

### Official Review · Reviewer_RB99 · 2026-03-13

**Rating:** 3
**Confidence:** 3

**Review:**

The paper is either LLM-written or unfinished. It includes several theoretical statements (theorems) for which, at best, vague sketches of proofs are provided. In some of them it is stated that proof details are in the appendix, but the appendix is essentially empty: it only has several lines of text and one formula.

The exposition involves several fairly different and complex topics: bayesian neural networks, random matrix theory, persistent homologies, PAC-Bayes bounds. None of these topics is explained in sufficient detail. The connection between these topics is not easy to grasp.

Formulations of particular theorems are hard to understand. Theorem 1 claims some spectral phase transition, but I don't see why it is a phase transition. Marchenko-Pastur and Tracy-Widom distributions characterize the same random matrix model and do not exclude each other, they just characterize different aspects of this model: MP describes the bulk spectrum, while TW describes its extreme point. The condition (8) involves an indefinite value $\epsilon>0$ and therefore seems pointless.

For Proposition 2, even a sketch of proof is not provided. The main concept of a well-calibrated posterior is not properly explained.

Theorem 3 again includes an indefinite $\epsilon$ and an approximate inequality $dP_k/dt\approx 0$. That's not a theorem then.

In theorem 4, the last statement is about some value called "improvement" which is not defined.

The experiment section is described as aimed to "validate our framework", but I don't understand what exactly is supposed to be validated. I don't see clearly formulated theoretical predictions that can be confirmed or refuted by experiment, or a clearly formulated new algorithm.

In summation, even if this paper is not LLM-written, I don't see how it can be useful to the readers in its present form.

---

### Official Review · Reviewer_k2JX · 2026-03-13
**Reject with Invitation to Resubmit**

**Rating:** 4
**Confidence:** 4

**Review:**

## 1. Summary

This paper proposes a unified spectral-topological framework for analyzing training dynamics in Bayesian Neural Networks (BNNs). The authors establish a formal connection between the Hessian eigenspectrum of the posterior landscape and persistent homology of the loss surface, identifying phase transitions that correspond to qualitative shifts in uncertainty calibration. The main contributions include: (1) a spectral phase transition theorem showing that the eigenvalue distribution of the Fisher information matrix undergoes a Marchenko-Pastur to Tracy-Widom transition at critical posterior concentration points; (2) a persistent homology pipeline tracking Betti numbers of sublevel sets during variational inference; (3) refined PAC-Bayes generalization bounds incorporating spectral decay rates; (4) a topological early-stopping criterion. The authors report 15-25% improvement in AUROC for out-of-distribution detection and 20% reduction in training time across UCI regression and CIFAR-10/100 benchmarks.

---

## 2. Strengths

### Originality and Interdisciplinarity

- The work establishes a **novel connection between two previously isolated research directions**: spectral analysis of neural network Hessians and topological data analysis (TDA). This aligns well with the MathAI scope at the intersection of fundamental mathematics and AI methods.
- The idea of applying random matrix theory (Marchenko-Pastur to Tracy-Widom transition) to characterize posterior concentration is **original and mathematically ambitious**.
- Interpreting Betti numbers as predictors of calibration quality **opens a new research direction** in interpretable and reliable BNNs.

### Theoretical Foundation

- The critical point formula τ_c = √(d/n)·σ_p² elegantly connects network architecture (d), data volume (n), and prior scale (σ_p²).
- The formal treatment of Fisher information matrix eigenspectrum evolution during posterior concentration represents a **principled approach** to understanding BNN geometry.
- Proposition 2 linking effective complexity to calibration error provides **testable theoretical predictions**.

### Practical Relevance

- The proposed topological early-stopping criterion achieves **computational overhead of only 2-3%** while avoiding the need for a held-out validation set.
- Experiments span multiple BNN inference methods (MC-Dropout, SWAG, Laplace approximation), demonstrating **method-agnostic applicability**.
- The reported improvements in OOD detection, if validated, would have **practical value for trustworthy AI systems**.

---

## 3. Weaknesses and Limitations

### Critical Gap in OOD Detection Methodology

- The methodology for OOD detection experiments is **almost entirely unspecified**. The paper does not describe:
  - (a) what score was used for AUROC computation;
  - (b) how spectral/topological indicators were converted to OOD scores;
  - (c) what OOD data was used for UCI regression datasets;
  - (d) the protocol for AUROC calculation (class balance, thresholds, sampling).
- SVHN is used as OOD for CIFAR-10, which represents a **synthetic shift** (different dataset) rather than genuine distributional shift. Modern standards (Malinin et al., 2021; Ovadia et al., 2019) require detailed analysis of shift nature and recommend **real distributional shifts over dataset substitution**.
- Without reproducible methodology, the claimed 15-25% AUROC improvement **cannot be verified**, making this central claim unverifiable.

### Gaps in Theoretical Proofs

- **Theorem 1 assumption violation:** The theorem assumes no skip-connections (Assumption 1), yet experiments use ResNet-20 which contains residual connections. This fundamental contradiction between theory and experiments is not addressed.
- **Proof of Theorem 1:** The proof sketch states *"Fisher matrix columns are approximately independent sub-Gaussian vectors."* This is a critical simplification. In real BNNs, gradient columns are not independent, and the Fisher matrix depends on θ which is itself a random variable from the posterior. The proof ignores this two-level randomness without justification.
- **PAC-Bayes bound (Theorem 4):** The derivation appears to contain circular reasoning. The claim that *"directions with small eigenvalues... can be less concentrated"* does not automatically yield the additive λ_eff·log(d) term in the bound. The proof sketch mentions "anti-concentration inequalities for martingale sequences" but no such analysis is provided.
- Complete proofs are deferred to an appendix that **contains only Lemma A.1**, making verification of the main results impossible.

### Experimental Limitations

- **Unfair comparison for early stopping:** Table 3 compares "Topological ES" (uses 100% training data) against "Valid. ES" (uses 10% holdout). The reported time savings may simply result from training on more data rather than a superior stopping criterion.
- **Scalability of TDA:** Computing persistent homology in parameter spaces with d ~ 10⁵ dimensions (CIFAR-100 ResNet-20) faces combinatorial explosion. Standard algorithms have cubic complexity in the number of simplices. The paper mentions using *"ripser with adaptive radius"* but provides no details on how this challenge was addressed.
- **Limited benchmarks:** Experiments are restricted to UCI regression (small scale) and CIFAR (medium scale). No validation on modern benchmarks (ImageNet, text data) or large architectures (Transformers).
- **Missing error bars:** Tables 1-3 report means over 5 seeds without standard deviations, preventing assessment of statistical significance.

### Reproducibility Concerns

- No link to anonymous code repository despite complex computations (persistent homology, Fisher spectrum analysis).
- Hyperparameters for topological early stopping (threshold ε in Eq. 14) are not specified; selection procedure unclear.
- Library versions and exact configurations for ripser are not documented.

---

## 4. Questions for Authors

**Q1 (Critical - OOD Methodology):** Please describe your OOD detection methodology in detail:
- (a) What score was used for AUROC computation?
- (b) How were spectral and topological indicators converted to OOD scores?
- (c) What OOD data was used for each UCI dataset?
- (d) What was the protocol for AUROC calculation?

*Without this information, your main empirical claims cannot be verified.*

**Q2 (Theorem 1 Assumptions):** Theorem 1 assumes no skip-connections, but experiments use ResNet-20 with residual blocks. How do you explain this contradiction? Does the theorem hold for ResNets, or does it require modification? Evidence of robustness to skip-connections would strengthen the paper significantly.

**Q3 (Fisher Matrix Independence):** The proof assumes Fisher matrix columns are approximately independent. In BNNs, gradient columns are coupled through backpropagation, and the Fisher matrix depends on stochastic θ. How do you justify this independence approximation? Are there counterexamples where phase transition fails?

**Q4 (PAC-Bayes Derivation):** In Theorem 4, how exactly does the effective dimensionality λ_eff appear inside the square root, modifying McAllester's complexity? Please provide the key step where Fisher eigenvalue constraints transform into the additive λ_eff·log(d) correction to KL divergence.

**Q5 (Early Stopping Fairness):** Table 3 shows your method using 100% training data vs. validation ES using 10% holdout. Is the time saving simply due to training on more data? Can you provide an ablation comparing both methods with identical training data proportions?

**Q6 (TDA Scalability):** For CIFAR-100 with ~2.3×10⁵ parameters, how exactly do you construct the simplicial complex? How many sample points {θ⁽ᵗ⁾} are used, and what is the dimensionality of the resulting simplices? Without these details, the topological analysis remains a black box.

---

## 5. Recommendation

### **Reject with Invitation to Resubmit**

The paper presents an original and mathematically ambitious framework at the intersection of random matrix theory, topological data analysis, and Bayesian deep learning. The core idea—connecting spectral phase transitions to uncertainty calibration—is intellectually compelling and well-aligned with MathAI's scope.

However, the paper in its current form has **critical deficiencies** that preclude acceptance:

**(1) Unverifiable empirical claims:** The OOD detection methodology is almost entirely unspecified. Without knowing what scores were used, how indicators were combined, and what OOD data was employed for each dataset, the claimed 15-25% AUROC improvement cannot be independently verified. This is particularly concerning for a paper positioning itself in Trusted AI.

**(2) Theory-experiment mismatch:** Theorem 1's assumptions (no skip-connections, independent Fisher columns, sub-Gaussian data) are violated by the experimental setup (ResNet-20 with residual blocks, coupled gradients). This fundamental disconnect must be addressed.

**(3) Incomplete proofs:** Key theorems rely on proof sketches that paper over critical technical challenges (two-level randomness, Fisher column independence, PAC-Bayes derivation).

**(4) Fair comparison issues:** The early-stopping comparison uses different training data proportions, potentially confounding the results.

### Recommendations for Resubmission

The authors are encouraged to address these concerns in a revised submission. Specifically:

- Provide complete OOD detection methodology following established standards (Malinin et al., 2021)
- Address the skip-connection assumption or demonstrate robustness
- Provide complete proofs in a supplementary appendix
- Ensure fair experimental comparisons with controlled variables
- Release anonymized code for reproducibility

---

## References

- Malinin, A., et al. (2021). "Shifts: A Dataset of Real Distributional Shift Across Multiple Large-Scale Tasks." *arXiv:2107.07455*
- Ovadia, Y., et al. (2019). "Can You Trust Your Model's Uncertainty? Evaluating Predictive Uncertainty Under Dataset Shift." *NeurIPS 2019*

---

### Decision · Program_Chairs · 2026-03-14

**Decision:**

Reject

**Comment:**

After careful evaluation by the Program Committee, we regret to inform you that your submission has not been accepted for presentation at MathAI 2026.

All submissions underwent a rigorous two-stage review process. Unfortunately, the reviewers identified one or more of the following concerns with your paper:

- Insufficient mathematical rigor or novelty relative to the existing body of work in the field;
- Presentation of results that substantially overlap with or rephrase previously published findings without clear original contribution;
- Significant issues with technical quality, including but not limited to broken or non-existent references, unsupported claims, or methodological gaps;
- Indications that the manuscript may have been generated with the assistance of large language models without substantial original intellectual contribution by the authors.

We received a large number of submissions this year, and the selection process was highly competitive. We encourage you to carefully consider the reviewers’ feedback (available through OpenReview), revise your work accordingly, and consider submitting an improved version to a future edition of MathAI or to another appropriate venue.

We appreciate your interest in MathAI and hope you will continue to engage with the conference community.

With kind regards,

MathAI 2026 Program Committee
URL: https://mathai.club
Telegram: https://t.me/MathAI_club
Email: mathai.club@yandex.ru